# Tracing the Extended Mind

Timothy Stanley

School of Humanities, Creative Industries and Social Sciences, The University of Newcastle, Callaghan, NSW 2308, Australia; timothy.stanley@newcastle.edu.au

**Abstract:** The following essay evaluates the concept of the trace within extended mind (EM) theory. It begins by differentiating Andy Clark's complementarity from several competing models. Second, it demonstrates how an undeveloped concept of the trace arises in Clark's debate with internalist critics. In response, I introduce Paul Ricoeur's metaphor of the imprint in *Memory, History, Forgetting*. Fourth, the recent debate about the plastic trace will be applied in this context. In so doing, the legacy of Jacques Derrida will be rehabilitated. I conclude with EM's renewed promise to model deliberations between religiously diverse people.

**Keywords:** extended mind; Andy Clark; complementarity; hermeneutics; Paul Ricoeur; Jean-Pierre Changeux; the trace; Jacques Derrida

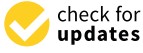

## 1. Introduction

In his 1997 *Being There*, Andy Clark depicts language as "just a tool—an external resource that complements but does not profoundly alter the brain's own basic modes of representation and computation" (Clark 1997, p. 198). Complementarity features throughout Clark's work, most notably in his account of the extended mind (Clark and Chalmers 1998, p. 18), and in essays that focus on language use (Clark 1998, p. 169; Clark [1998] 1999, p. 13; 2006a, p. 371; 2006b, p. 291). Delimiting language as a tool may seem to diminish the importance of its contribution to cognition. However, Clark contends that the opposite is the case. Language is "special" precisely because it is "productively poised between the inner and the outer, the private and the public, the biological and the artifactual" (Clark 2004, p. 725). Nowhere is this complementarity more significant than when it impacts moral and religious deliberation. In *Supersizing the Mind*, Clark points to several of his previous essays about how language complements moral reasoning (Clark 2008, p. 53, Cf.; 2000a, 2000b, 1996). In each case, he argues that "moral expertise" relies on "collaboration and reason made available by the tools of words and discourse" (Clark 2000b, p. 269, Cf.; 2000a, p. 307; 1996, p. 122). In sum, language uniquely contributes to "more basic forms of neural processing" by complementing the human capacity to solve problems, not least of which, when facing disagreements in deliberative encounters (Clark 2008, p. 53).

Complementarity captures key differences "across brain, body, world, and artifact" (Clark 1997, p. 218). When considering how language features across those differences, Clark acknowledges that "the inner goings-on involve, as genuinely constitutive elements, something like images or traces of the public language symbols (words) themselves" (Clark 2006b, p. 301). My contention is that the trace is an undeveloped concept in EM that calls for further evaluation. The following essay begins by differentiating Clark's complementarity from several competing models. Second, it demonstrates how the trace arises in Clark's debate with internalist critics. In response, it introduces Paul Ricoeur's

metaphor of the imprint in *Memory, History, Forgetting*. Fourth, the recent debate about the plastic trace will be evaluated in this context. In so doing, the legacy of Jacques Derrida will be rehabilitated. I conclude with EM's renewed promise to model deliberations between religiously diverse people.

## 2. Clark's Complementarity

There are several alternatives that can help illuminate the distinctiveness of Clark's complementarity. For instance, he cites Peter Carruthers's *Language, Thought and Consciousness*, which provides an "extended debate with Jerry Fodor", who viewed inner thought as an "innate, universal, symbolic system" or form of "mentalese" (Carruthers 1996, p. 4, Cf.; Fodor 1994, p. 105ff). In a later collection of essays on *Language and Thought*, Clark recognizes that Carruthers is quite close to his own view (Clark 1998, p. 166). Clark focuses on one of Carruthers' examples of thinking with a notebook. As Carruthers has it, "One does not *first* entertain a private thought and *then* write it down: rather, the thinking *is* the writing" (Carruthers 1996, p. 52, cited in; Clark 1998, p. 166). This might seem to be an example of "The Extended Mind" that Clark published in January of that same year with David Chalmers. One of the examples there also applied the use of notebooks to extend the minds of those with memory impairment (Clark and Chalmers 1998, pp. 12–13). To be clear, Carruthers is not arguing that language changes the brain by reprogramming it akin to Daniel Dennett's view in *Consciousness Explained* (Dennett 1991, p. 278). Carruthers is also "careful to reject what he calls the 'Whorfian relativism of the Standard Social Science Model'" (Clark 1998, p. 166, citing; Carruthers 1996, p. 278, Cf.; Whorf 1956). Rather than the transformation of thought, "language is constitutive of much of human conscious thinking, in much the same sense that water is constituted by $H_2O$" (Carruthers 1996, p. 278). Conceptual diversity across different languages calls for careful investigation. Such studies are established on scientific grounds as "a crucial, and central aspect of the human mind" (Carruthers 1996, p. 278). Carruthers acknowledges that he does not prove this conclusion so much as to set it out as a plausible account for further neuroscientific research.

However, for Clark, the question is not "do we actually think *in* words" but rather how do words benefit "biological pattern-completing brains" (Clark 1998, p. 169). Said another way, establishing an essential identity between the brain and inner speech is a moot point that distracts from Clark's aim to study features of "the extended mind" (Clark 1998, p. 179). In EM, it is not the brain that is transformed so much as the problems themselves "into formats better situated to the perceptual and pattern-completing capacities of the biological brains" (Clark 1998, p. 169). For Clark, the models matter as much for future neuroscientific research as for the development of artificial intelligence. Both depend on surpassing the misguided "gross separation between the biological agent and an external scrolling of ideas persisting on paper, in filing cabinets and in electronic media" (Clark 1998, pp. 180–81). Biological brains are better understood as augmented. In *Natural Born Cyborgs*, Clark more pointedly argues that "what blinds us to our own increasingly cyborg nature is an ancient western prejudice—the tendency to think of the mind as so deeply special as to be distinct from the rest of the natural order" (Clark 2003, p. 26). In contrast, the mind is best modeled as extended to the outer environment as a cyborg or cybernetic organism that includes a "human–machine merging" (Clark 2003, p. 14). Here again, however, it is crucial to note that cybernetics captures the complementary communicative connection between the organism and the various tools they may use.

In "Material Symbols", Clark returns to Jerry Fodor's "mentalese" with the aim to clarify its relation to his own "'complementarity' view of language" (Clark 2006b, p. 291, Cf.; Fodor 1998, p. 67). Clark critiques what he perceives to be a dualism that emerges between language and inner thought in Fodor's work. "Language impacts thought, on

such accounts, in virtue of a process of translation that transforms the public sentence into a content-capturing inner code" (Clark 2006b, p. 292). Translation models of language occur in other forms, and Clark cites Paul Churchland's "neuralese" as another example (Clark 2006b, p. 300). In *A Neurocomputational Perspective*, Churchland argues that public language provides a "one dimensional projection of a four- or five-dimensional solid that is an element in his true kinematical state" (Churchland 1989, p. 18). However, for Clark, both Fodor and Churchland share in this translational model's error in depicting public language as secondary to inner mental states (Clark 2006b, p. 300). By contrast, complementarity emphasizes the way language straddles "the internal-external borderline itself, looking one moment like any other piece of the biological equipment, and at the next like a peculiarly potent piece of external cognitive scaffolding" (Clark 2006b, p. 293). Fodor and Churchland's error persists, irrespective of whether consciousness or inner mental representations are under consideration. However, Clark also recognizes that Fodor is not inherently opposed to an alternative model of extended language (Clark 2006b, p. 301, citing; Fodor 1998, p. 72).

As Clark points out, Fodor's account of the language of thought is unconcerned if it turns out that an inner mentalese "co-opts bits of natural language" (Fodor 1998, p. 72). But this assumes that "the basic biological engine... comes factory-primed with innovations favoring structure, integration, generality and compositionality" (Clark 2006b, p. 301). However, if the brain does not already have a notion of mentalese of this kind in place, then the consequence of language's contribution to the brain becomes more important. While Fodor is open to "hybridity", Clark's understanding of how language complements the brain differs from Fodor's by removing the need for a full-fledged inner language of thought (Clark 2006b, p. 300). Clark does not explain how precisely this co-opting takes place. He is also ambiguous at times about how "conscious and unconscious access to representations of language-specific lexical items" occurs (Clark 2006b, p. 296). He recognizes that the precise mechanism is elusive and again focuses on examples of problem-solving systems such as simulated agents that can re-use a "public symbol system to aid cognition" (Clark 2006b, p. 302). When considering writing an essay, Clark acknowledges all the notes and bits of texts upon which he relies. The final product is not just a matter of inner cogitation. "Instead, it is the product of a sustained and iterated sequence of interactions between my brain and a variety of external props" (Clark 1998, p. 176). An integrated view of the human being as a biological system can arise by depicting such practices without mentalese. EM for Clark is better understood as a mangrove where seeds sometimes grow into island chains (Clark 1998, p. 176). "It is natural to suppose that words are always rooted in the fertile soil of pre-existing thoughts. But sometimes, at least, the influence seems to run in the other direction" (Clark 1998, p. 176). Here again, the mangrove metaphor maintains a distinction between brain and language that leaves open the nature of the symbiosis.

Importantly, this "mangrove effect" not only more accurately reflects the extended nature of human minds, but it also has profound implications for problem solving amidst moral and religious disagreement. In an essay on "Connectionism, Moral Cognition, and Collaborative Problem Solving", Clark considers "the case of a conflict within a multicultural educational system" (Clark 1996, p. 122). In such situations, he argues, moral principles "attempt to lay out some rough guides and signposts that constrain the space to be explored in the search for a cooperative solution" (Clark 1996, p. 122). While disagreement about moral rules appears to be the source of the conflict, Clark suggests they are better understood as examples of "expertise" in orchestrating a "practical solution sensitive to multiple needs and perspectives" (Clark 1996, p. 122). At this point, he draws on Jürgen Habermas's distinction between strategic and communicative action to note how

moral expressions are not just about persuading others to agree. Rather, they also include mutual recognition as individuals pursue "a dialogue by visibly committing oneself to a *negotiated* solution" (Clark 1996, pp. 122–23, citing; Habermas 1990, pp. 58, 59, 134, 145). Moral language is thus reframed as a collaborative medium, neither the "heart of moral reason" nor just a "distortive gloss" on connectionist accounts of cognition (Clark 1996, p. 109, citing; Churchland 1989). Moral knowledge relies on "concrete examples of moral judgments" rather than abstract principles divorced from social contexts. It therefore "may not be fully reconstructible in the linguistic space afforded by public language" (Clark 1996, p. 114). Again, this assumes that language complements the brain's experiences in ways that it could not achieve on its own. Our "wetware" cannot express its moral interests in such a way that communication can meaningfully occur, which is where "wideware" such as language comes into its own (Clark [1998] 2008, pp. 2–3, 10). To attempt to represent all moral experience in a few principles may be as inadequately optimistic as trying to "reduce a dog's olfactory skills to a small body of prose" (Clark 1996, p. 114). Precisely for these reasons, Clark reiterates the collaborative problem-solving importance of moral language. It provides one of the ways we come to express a moral principle that can be understood and evaluated by others.

Clark's focus on moral disagreement conceals a religious concern. The case he provides regards the parents of a Muslim girl who requested that she be "excused from events involving what (from their perspective) was an unacceptably close physical proximity to boys" (Clark 1996, p. 122). For Clark, the only chance at a solution that did not result in the girl leaving school relied on "attempts by each party to articulate the basic principles and moral maxims that inform their perspective" (Clark 1996, p. 122). Again, the goal is not agreement so much as an empathetic encounter. However, coordinating religious disagreement relies on the special contribution language makes to collaborations between diverse people. This aligns with the two main themes Clark uses to depict language's many practical complements to cognition, firstly as "forms of external memory", and secondly as "transformers of the very shape of the cognitive and computational spaces we inhabit" (Clark 1998, p. 174). Clark assumes that language transforms opaque opposition into the potential to recognize the viewpoints of others (Clark 1996, p. 125). However, this obscures the challenges that have been cited about the difficulty of coordinating religious disagreement in a world increasingly shaped by religious strife.

Habermas has since recognized the ongoing challenge of religious tolerance, which he described at times as a "pacemaker for multiculturalism" (Habermas 2004, p. 9). I have argued elsewhere, however, that solutions need not be limited by Habermas's approach to universal discourse (Stanley 2022b, p. 12). According to Jeffrey Stout, ethical deliberations can involve richly contextualized pidgins where the translation of diverse religious commitments takes place (Stout [1988] 2001, pp. 79–81; 2004, pp. 97–98). As he put it, "the 'sad little joke' about universal languages, Mary Midgley once said, is that almost nobody speaks them" (Stout [1988] 2001, p. 166, citing; Midgley 1978, p. 306). In the case of religious discourse, there are further considerations given different languages such as Arabic and references to sacred texts require great skill to interpret and translate in their many contexts. However, as historians of religion such as J. Z. Smith have suggested, while translation practices are never complete in the religious case, they nonetheless provide models and maps of the subject matter that require special care (Smith [2001] 2004, pp. 371–72, Cf.; Stanley 2022a, p. 41). A translation's efficacy is evaluated according to its ability to map a given territory (Smith 1978). My contention is that Clark's complementarity adds a cognitive dimension that can enhance models of religious deliberation in diverse democracies, a point I will return to in my conclusions below.

Cognitive science of religion scholars have recommended EM given its emphasis upon "connections between the body, the brain, consciousness, language, culture and religion" (Jensen 2013, p. 248, citing; Clark 1997). However, past applications of EM to the study of religion have typically overlooked complementarity's importance. Rather, they have aimed to apply EM to explain the "computational challenge of religious cognition" (Day 2004, p. 109, Cf.; Krueger 2017, p. 243), and to explain how "layers of cognitive technology" make the gods thinkable. EM has also been applied to religious rituals and intersubjective ideas like empathy (Schilbrack 2014, p. 46; Krueger 2009, p. 690). EM thus inaugurates new approaches to the study of material religious culture. In so doing, these applications of EM to religion tend to focus on how an internal process and an external notebook "constitute a single cognitive system" (Clark and Chalmers 1998, p. 16). Hence, Krueger discusses how "some mental states and processes are partially constituted by external resources" (Krueger 2017, p. 240). He also notes that "external architecture... plays a constitutive role in shaping the 'inner' subjective experience of emotional states" (Krueger 2009, p. 687). Day similarly blurs the distinction between language and biological brain activity in Clark's complementarity, at one point depicting religious artifacts as "the media of thought" (Day 2004, p. 117). Day cites Clark's essay on the importance of moral reasoning (Day 2004, p. 114n10, Cf.; Clark 1996, p. 124), but he remains focused on how religious material culture in "rituals, music, relics, scriptures, statues and buildings... begin to look like central components of the relevant machinery of religious thought" (Day 2004, p. 116). It is true that Clark's mangrove effect encompasses an overall cognitive system. However, Clark's complementarity also aimed to apprehend differences between the brain and external artifacts. Each makes distinct contributions that must be adequately appreciated on their own terms. The counterpoint noted throughout Clark's work above is that public language does not constitute inner thought and cannot be reduced to brain activity.

## 3. Parity's Trace

Part of Clark's challenge is that complementarity does not identify the precise mechanisms of traffic between brain activity and public language. As he concludes at one point, "It would be good to have much more in the way of genuine, implementable, fully mechanistic models of the various ways that internalized language might enhance thought" (Clark 2006b, p. 304). This has led to significant debate about EM's parity principle as a potential source of that mechanism. In "The Extended Mind", four features were summarized according to a notebook's constancy and availability. In addition to that, the notebook's information needed to be endorsed by its user, Otto, in the present and past (Clark and Chalmers 1998, p. 17). Clark and Chalmers both have clarified that parity relies on the function of different parts of problem-solving systems where the brain is complemented by external artifacts (Chalmers 2008, p. xv; Wilson and Clark 2009, p. 69). They also noted how the "physical traces" found in notebooks should be treated "as the physical vehicles of some of Otto's own non-conscious mental states" (Wilson and Clark 2009, p. 67). Clark does not develop this concept at this point, but in *Supersizing the Mind*, he again references the traces found in both material notebooks and "internal memory traces" (Clark 2008, p. 76).

Critics have focused on the validity and scientific utility of parity, favoring instead the internalist view Clark cites as a prejudice above (Adams and Aizawa 2008, 2009, 2010). This concern is summarized in Adams and Aizawa's essay on "Why the Mind Is Still in the Head", which takes issue with what counts as "cognitive" in EM (Adams and Aizawa 2009, p. 80). When considering a particular example of tool use, such as the pen and paper when writing an essay, they do not see an example of EM, but rather the "coupling-constitution fallacy" (Adams and Aizawa 2009, p. 83). There is no reason "to think that the tools and the brain constitute a single 'cognitive' process" (Adams and Aizawa 2009, p. 83). To suggest

that tool use can be depicted as a "cognitive system" does not result inevitably in the view that this tool use can "constitute transcranialism" (Adams and Aizawa 2009, p. 84). Much, therefore, comes down to the meaning of what constitutes the brain's cognition against its environment. As Adams and Aizawa note, however, "Clark and Chalmers, like all the other transcranialists we have read, do not address any version of the rules-and-representations conception of cognition—arguably the received view of the nature of cognition (Adams and Aizawa 2009, p. 87). Adams and Aizawa thus aim to locate the model of the brain as the source of their critique of EM.

Precisely here, Clark and Wilson cite this impasse as the key. In an essay from the same collection as Adams and Aizawa's critique, they take issue with their claim that EM's "vision of the inner realm departs fundamentally from that of classical cognitive science" (Wilson and Clark 2009, p. 72). As noted above, Clark's view is that "linguistic symbols and encodings contrast dramatically with the fluid, distributed, context-sensitive representations developed by a connectionist or dynamical engine" (Wilson and Clark 2009, p. 73). Hence, the case of EM notebooks "depends not on fine-grained functional identity but on the deep complementarity of inner and outer contributions" (Wilson and Clark 2009, p. 71). To drive this point further, they contrast the hypotheses of "in-brain cognition" with "in-neuron cognition" (Wilson and Clark 2009, p. 74). "Our best empirical research tells us that intuitively cognitive acts often involve lots of neurons spread throughout the brain" (Wilson and Clark 2009, p. 74). It, therefore, makes little sense to pose the interior, however small or focused that can be defined, as the truer mark of the cognitive. Rather, the response to brain science as we have it should adapt to accurately depict the connections. In "Material Symbols", Clark noted, "by stressing coordination dynamics and hybrid representational forms, we leave room for genuine complementarity between the biological and artifactual cognitive contributions" (Clark 2006b, p. 300). The pursuit of a pure inner, inner representational mark of cognition obscures the various ways public artifacts complement the biological brain.

Given that part of Clark and Wilson's rebuttal relies on Churchland's connectionism, one might presume that he would be sympathetic to Clark's EM. However, when it comes to the value of moral artifacts, Churchland also opposes public moral codes to an inner "scientific cognition" (Churchland 2000, p. 291). In this, Churchland continues his longstanding opposition between "cognitive and behavioral *skills*" and "a certain set of *rules*" evident in his earlier work (Churchland 1996, pp. 292–93). For similar reasons to his critique of Adams and Aizawa, Clark rejects Churchland's opposition as a "mistake" in favor of his complementarity between the brain and public moral deliberation (Clark 2000b, p. 291). However, Clark argues that this impasse is the result of moral rather than scientific reasons, which means it cannot be resolved by neuroscience itself (Clark 2000a, p. 312). However, one of the central features of EM concerns its capacity to bridge these divides. Proponents of EM, such as John Sutton, have recognized its promise for studying "cognition scientifically and culturally at once" (Sutton 2010, p. 215).

In part, the disagreement between Clark and his critics persists because of how the concept of constitution features in EM's account of the cognitive system. Other commentators have clarified this point by noting that "the case of extended mind doesn't rest on such a case for functional isomorphism between the inner and the outer" (Sutton 2004, p. 210). Hence, the concept of constitution is typically reserved for the mark of the system, which Clark envisions in a connectionist manner. To combat the persistent argument for depicting the constitution of inner states, Clark suggests that the pursuit of the inner results in an infinite regress. At this point, he cites a similar concern in Fodor's responses to those who saw, in his *The Language of Thought*, "an endless regress of such codes" (Clark 2006b, p. 300, Cf.; Fodor 1975, p. 65). Fodor responded by claiming that what he would later

call mentalese must be "innate" (Fodor 1975, p. 65). However, Clark is not committed to a notion of mentalese so much as the meaningful complementarity between public natural language and inner brain activity. Complementarity seems to act in the interplay of connectionist differences.

In response, Clark emphasizes "the potential for representational hybridity as massively important to understanding the nature and power of much distinctively human cognition" (Clark 2006b, p. 301). At this point, he relies on the concept of a trace in the "biological engine" (Clark 2006b, p. 301). It is a passing remark, but a longer quotation from this section of "Material Symbols" makes the case.

> The potential cognitive impact of a little hybridity and co-opting may be... essential to such a system's ability to think rather a wide variety of thoughts that the inner goings-on involve, as genuinely constitutive elements, something like images or traces of the public language symbols (words) themselves. Words and sentences, on this view, may be potent structures many of whose features and properties (arbitrary amodal nature, extreme compactness and abstraction, compositional structure, and so on) deeply complement the contributions of basic biological cognition. (Clark 2006b, p. 301)

Complementarity here includes a constitutive relationship between inner and outer traces. Clark does not develop this concept at this point, but in *Supersizing the Mind*, his account of inner traces relies on a distinction between vehicles and content (Clark 2008, p. 76). At this point, Clark notes how the content of mental states is not the same as the material vehicles that produce them. A trace includes both aspects, whether it is operative in a notebook or in the brain. This distinction aims to defend EM against concerns that the content of a mental state is the same as a record in a notebook. Hence, EM "is really a hypothesis about extended vehicles—vehicles that may be distributed across brain, body, and world" (Clark 2008, p. 76). Moreover, the vehicles can also differ from each other so long as they each contribute to the overall system in the production of dispositional beliefs. "This occurs if the traces become poised for the control of action in roughly... the same kind of way as internal memory traces" (Clark 2008, p. 76). This maintains complementarity, given that written traces differ from "the fluid, automatically responsive resources of internal biological memory" (Clark 2008, p. 77). However, the main difference Clark delineates at this point is between the material traces as vehicles, which should not be confused with the dispositions believed in mental states.

Clark cites Daniel Dennett's *Consciousness Explained* in support of the vehicle/content distinction. However, it is worth noting that Dennett's elaboration on this point distinguished "representing from represented, vehicle from content" (Dennett 1991, p. 131). Clark largely succeeds in maintaining differences between vehicle traces that again remain different from mental states. In so doing, however, the trace leaves several matters unresolved that I will address in the next two sections. First, Clark considerably underestimates the representational complexity of the trace. Traces can misrepresent and, in part, this is because they operate in the absence of past events. Second, it remains unclear how the different vehicle traces, whether internal or external, relate to each other in the production of mental states. It may be that the precise mechanisms in the brain will remain unknown for some time. However, the trace's differences call for an analysis of its potential generality. The first aspect has been magisterially addressed in Paul Ricoeur's *Memory, History, Forgetting* (Ricoeur 2004). The second appears in new materialist accounts of the plastic trace.

## 4. The Paradoxical Trace

In one of his final works, Ricoeur argued that the "correlation between neurology and phenomenology is equivalent to a correspondence", and he cited Clark's *Being There*, as a prominent example (Ricoeur 2004, pp. 422 and 591n9, citing; Clark 1997). It should be clear from the above that Clark's complementarity encompasses practical considerations of neuroscientific examples that go beyond Ricoeur's interests. Nonetheless, Ricoeur's study of the trace is worth rehearsing, given their shared concern for a similarity between brain and mental experience that does not reduce it to an identity. As Ricoeur summarized, the trace has three aspects, "the written trace… the psychical trace… [and] the cerebral, cortical trace, which the neurosciences deal with" (Ricoeur 2004, p. 415). This tripartite distinction encompasses EM's interest in written artifacts, language use, and the biological brain. However, Ricoeur argues that the trace results in a paradox that goes back to Plato and which continues to challenge any simple reduction of the mind to neuroscience (Ricoeur 2004, p. 15). In part, this is because the trace itself represents the absence of past events.

For instance, in the *Theaetetus*, Socrates asked his interlocutor to assume, for argument's sake, "that there is in our souls a block of wax" (Plato 1921, p. 185/191). For this reason, memories can be considered as imprints of varying degrees of quality. "Whenever we wish to remember anything we see or hear or think of in our own minds", Socrates avers, "we hold this wax under the perceptions and thoughts and imprint them upon it" (Plato 1921, p. 185/191). There is an empirical worry here as Plato would similarly argue in the *Sophist*, where he "distinguishes veracity from trickery in the order of imitation" (Ricoeur 2004, p. 11, Cf.; Plato 1921, p. 329/234). How, in other words, do we know whether the imprint is true to the event? In Ricoeur's summation, Plato introduced skepticism about material representation, whether in an imagined interior wax, a work of art, or in the mnemonic techniques of memory (Ricoeur 2004, pp. 11–13, Cf.; Krell 1990, pp. 25–28). This skepticism is echoed in other works, such as *Phaedrus*, where writing's invention is similarly critiqued as a danger to memory (Plato 1995, p. 563/275). Plato's response ultimately culminated in the allegory of the cave, where material shadows are contrasted with the illumination provided by the light of the sun (Plato 1935, p. 119, Book VII, 515c). In *The Republic*, Plato reflected upon the truer realm of ideas (*eidos*) and, ultimately, "the idea of the good" (Plato 1935, p. 131, Book VII, 517c). As commentators have noted, Plato has thus inspired several varieties of "transcendental metaphysics" up to the present day (Shorey 1935, p. xxix).

Ricoeur was careful to avoid several difficulties in Plato's example. First, he did not fall back on the transcendental implications of Platonic idealism. He was, therefore, vigilant against a notion of the mind as "immaterial" (Ricoeur 2004, p. 419). Second, he did not displace the sunlit exit of the cave with the promise of neuroscience (Churchland 1989, p. 18). Rather, he aimed to include the body in a way that is "irreducible to the objective body as it is known in the natural sciences" (Ricoeur 2004, p. 419). Third, Ricoeur was wary of reiterating a substance dualism between mind and body. The problem that he identified in Plato's metaphor of wax in the *Theaetetus* is that it established "a dialectic of presence, absence, and distance that is the mark of the mnemonic phenomenon" (Ricoeur 2004, p. 426). It is important to note that Ricoeur contrasts mnemonic experience with mnestic or cortical traces in the neurosciences (Ricoeur 2004, p. 590n3). For Ricoeur, the mnemic trace can only be analyzed at the level of the conscious experience of memory. As Ricoeur put it more pointedly, "only discourse about the mind can account for this dialectic" (Ricoeur 2004, p. 426). Hence, despite Plato's own philosophical interests, his metaphor continues to draw attention to the problem of how to "make a function correspond to an organization" (Ricoeur 2004, p. 426). The neuroscientific task is, therefore, *not* to explain what makes me think. All it can do is account for "the neural structure without which I could not think" (Ricoeur 2004, p. 426). This is not to diminish the brain's importance for

Ricoeur. Rather, like Clark, Ricoeur thoroughly differentiates the brain from the mind. To this end, Ricoeur cites the trace as a crucial metaphor that structures that correlation.

In his debate with Jean-Pierre Changeux in *What Makes Us Think?*, Ricoeur aimed to explain how the trace arose in neuroscientific studies of memory. As he argued there, the trace was introduced "in order to account for exactly this presence of something absent" (Changeux and Ricoeur 2000, p. 146). Changeux agreed by noting two progenitors who studied memory traces quantitatively through various measures of the rate of forgetting meaningless stories and syllables (Changeux and Ricoeur 2000, p. 147, Cf.; Bartlett 1932; Ebbinghaus 1913). However, for Ricoeur, such studies on the brain's contribution to memory only reiterate the problem. It does not matter whether the trace makes its imprint in Socratic wax or any other depiction of neuronal activity the neurosciences discover. The trace forces a question about the "relation between the neuronal basis of mental experience and the experience itself" (Changeux and Ricoeur 2000, p. 150). Hence, Ricoeur leaves "open" the question of how language and memory are connected at this point (Changeux and Ricoeur 2000, p. 145). The relation between function and organization does not rely on resolving that issue. Even if it turns out we think in images as Changeux surmised, "the hieroglyphs would still have to be deciphered, as when the age of a tree is read by counting the concentric circles drawn on the tree stump" (Ricoeur 2004, p. 426). The immediacy of the image cannot resolve how it is possible to interpret them in human experience.

As Ricoeur put it more pointedly, "there is no parallel between the two sentences: 'I grasp with my hands', 'I understand with my brain'" (Ricoeur 2004, p. 420). Even as the trace encompasses a written artifact and a cortical activity, it does not contravene this basic problem. We only have access to our brains through scans or through language that we draw upon to interpret the past. Ricoeur was as concerned as Plato and the neurosciences about the "trustworthiness of memory" (Changeux and Ricoeur 2000, p. 150). However, he concentrated upon the semiotic dimension of the trace precisely to focus on the problem of how function and organization relate. Hence, when returning to this issue in *Memory, History, Forgetting*, he cited again his debate with Changeux to emphasize the point that "it is necessary then to endow the trace with a semiotic dimension, so that it functions as a sign, and to regard the trace as a sign-effect, a sign of the action of the seal in creating the impression" (Ricoeur 2004, p. 425, citing; Changeux and Ricoeur 2000, p. 149). The semiotic dimension of the trace is its action as an imprint in the face of absence. As he concluded to Changeux, "a trace must therefore be conceived at once as a present effect and as the sign of its absent cause" (Ricoeur 2004, p. 426, citing; Changeux and Ricoeur 2000, p. 150).

The "metaphor of the imprint does not resolve the enigma of the representation of absence and distance" (Ricoeur 2004, p. 426). Ricoeur does not go into detail about the role of metaphor at this point. However, it is worth noting how he criticized the vehicle/tenor distinction in *The Rule of Metaphor* (Ricoeur 1977, p. 65, citing; Richards 1936, p. 96). It depicts a sign's surface as a vehicle for its underlying meaning or tenor, making it cognate to Clark's vehicle/content distinction noted above (Ricoeur 1977, p. 93). The problem is that this distinction cannot resolve the tension between a literal and figurative interpretation that metaphor demands. Metaphorical interpretation exceeds the attempt to map a signifier to a signified (Ricoeur 1977, p. 94). Instead, Ricoeur argued that metaphors rely on the "self-destruction" of literal interpretations (Ricoeur 1977, p. 271). Focus on the vehicle/tenor distinction obscures this activity, to which Ricoeur's hermeneutic analysis responds. A similar strategy is required to apprehend Ricoeur's analysis of the trace. The problem is not that there is a trace that corresponds to a dog one might see in a park and pencil it into a notebook. For Ricoeur, there is a semiotic dimension to a dog's multiple meanings (e.g., a poodle, a sausage, etc.) that can only be resolved in a semantic context. However, in light of the metaphor of the imprint, the trace represents the imprinting action

itself. The question of how function and organization relate surpasses the limits of simply mapping representation to what is represented. The metaphor of the imprint calls for a new understanding.

In pursuit of that meaning, Ricoeur cites an unresolved opposition between a "destructive forgetting" and a "forgetting that preserves" (Ricoeur 2004, p. 442). Ricoeur acknowledged the paradox of the latter case, or as he put it, "forgetting makes memory possible" (Ricoeur 2004, p. 442). He located this paradox in Martin Heidegger's choice of terms when describing the past in *Being and Time*. Instead of the everyday German term "*Vergangenheit*", he preferred "*Gewesenheit* (having-been)" (Ricoeur 2004, p. 442; Heidegger 1962, pp. 373–74/326). The former indicated a "past that has expired" or events subject to an "inexorable destruction" (Ricoeur 2004, pp. 442–43). In contrast, "having-been makes forgetting the immemorial resource offered to the work of remembering" (Ricoeur 2004, p. 443). For these reasons, Heidegger designated "temporality (*Zeitlichkeit*)" as "the unity of a future which makes present in the process of having been" (Heidegger 1962, p. 374/326). For Ricoeur, only through temporality can we analyze memory given it appears to us in this sense, "at hand (*zuhanden*)" (Ricoeur 2004, p. 442). Here, the difference between an absolute past and a past that is available to us is recalled in Heidegger's differentiation between a tool's material makeup or presence at hand (*Vorhandenheit*) versus its use as something ready to hand (*Zuhandenheit*) in a carpenter's workshop (Heidegger 1962, pp. 114–15). Ricoeur's analysis of Heidegger's paradox uncovered a past that is at hand precisely in its absence.

In the end, Ricoeur did not resolve this paradox. It rather exhausted his analysis of the correspondence between brain and mind as "fundamentally undecidable" (Ricoeur 2004, p. 443). Ricoeur thus encouraged dialectics to resist any simple reduction of thinking to neuroscientific inquiry. This gap is consistent with his view of the philosopher's role "to relate the science of mnestic traces to the problematic central to phenomenology, the representation of the past" (Ricoeur 2004, p. 419). In this sense, his trace respects aspects of Clark's complementarity. What Ricoeur brings into focus is how the trace's representational activity revolves around the absence of the past. When he considered the "cortical trace", he would also do so with reference to a paradox that arises from "the dialectic of presence, absence, and distance" (Ricoeur 2004, pp. 418–19). This activity does not explain how mental states arise from cortical traces so much as to delimit their role. In so doing, Ricoeur established hermeneutics as the primary mode to study written and linguistic traces, whose operations are more clearly at hand (*zuhand*). For these reasons, Ricoeur's debate with Changeux has appeared to some critics as the persistent opposition "between reductionism and antireductionism" (Malabou 2008, p. 81). In response, they explore the generality of the plastic trace as a logical consequence of its material instantiations.

## 5. The Plastic Trace

The persistent gap between philosophy and neuroscience has led Catherine Malabou to recommend new concepts such as "plasticity" in response (Malabou 2008, p. 82). However, she also opposes plasticity to terms inevitably tied to metaphors of graphic writing, such as the trace. Like Ricoeur and Clark, Malabou acknowledges that it is not "defensible to advocate an absolute transparency of the neuronal in the mental, an easy back-and-forth from the one to the other" (Malabou 2008, p. 82). Following Changeux, she argues that "the synapse is the privileged locus where nerve activity can leave a trace that can displace itself, modify itself, and transform itself through repetition of a past function" (Malabou 2008, p. 22). Rather than develop this notion of the trace, Malabou opposes it with plasticity to better capture the brain's flexibility. "Plasticity designates solidity as much as suppleness, designates the definitive character of the imprint, of configuration, or of

modification" (Malabou 2008, p. 15). This seems to align her philosophy with the plasticity that appears in Changeux's depiction of "human cerebral organization", in *What Makes Us Think?* (Changeux and Ricoeur 2000, p. 152). However, the difficulty is that Changeux also uses the trace to describe what appears in brain imaging techniques (Malabou 2007, p. 440). In Changeux's words, in brain scans, "we have at our disposal physical traces of how meaning is accessed" (Changeux and Ricoeur 2000, p. 107). Malabou suggests that "the 'traces' of which Changeux speaks here are, in fact, first and foremost images and forms" (Malabou 2007, p. 440). The graphs that display after brain scans are not written in Malabou's view because they represent neuronal "*assemblies*, of *formations* or of *neuronal populations*" (Malabou 2007, p. 440). Plasticity aims to apprehend this neuronal behavior in a way where writing metaphors mislead by depicting fixity and permanence.

There remains a proximity between the trace and plasticity in Malabou's analysis. However, she, in part, denies this connection for reasons related to her opposition to Ricoeur and even more so to Jacques Derrida's grammatology (Malabou 2007, p. 431). In *Of Grammatology*, Derrida outlined the scope of this new "science of writing" (Derrida [1967] 1978, p. 4). However, Malabou argues that such a science "has never existed" (Malabou 2007, p. 431). Moreover, she outlined how Derrida's "grammatology cannot be a science like other sciences" (Malabou 2007, p. 433). Hence, "plastology" aims to better align philosophical reflection with neuroscientific developments (Malabou 2007, p. 439). Commentators have since questioned the degree to which this break with the trace is necessary, especially given its continued application by neuroscientists such as Changeux and Clark, as noted above. This in no way undermines Malabou's contribution. Rather, as Deborah Goldgaber has argued, it draws attention to the need to consider a "plastic trace" (Goldgaber 2020, p. 153). When Malabou speaks of a "plastic coding of experience", she draws upon neuroscientists who present their work in textual terms (Goldgaber 2020, p. 153). She also notes that several of the examples Malabou cites are not settled science, such as that of Phineas Gage, "the nineteenth-century railway worker whose skull was impaled by an iron spike" (Goldgaber 2020, p. 161, Cf.; Johnston and Malabou 2013, pp. 57–58). The problem is that depicting plasticity in this example overlooks how "there is absolutely no consensus (either among historians of science or scientists) about what sort of cerebral transformations Gage underwent" (Goldgaber 2020, p. 162). In response, Goldgaber provides reason to think that the trace continues to be relevant, precisely because it includes a plastic inflection. Malabou includes the potential for this connection when she cites plasticity's "imprint" (Malabou 2008, p. 15). For these reasons, Goldgaber cites the persistent relevance of Derrida's account of the trace in this context. Similarly, others have echoed the potential to open "his philosophy to a future, which is not his own" (Crockett 2018, p. 120).

While Derrida did not develop a science of writing, he was aware of grammatology's empirical implications. In other contexts, he explicitly affirmed "the necessity of scientific work in the classical sense" (Derrida 1970, p. 271). Derrida did not provide an empirical science of writing in *Of Grammatology*, but he *did* outline how such a science might proceed and remained open to its application to scientific fields. This explains why it inspired several projects such as *Applied Grammatology* and *Cultural Graphology* (Fleming 2016, Loc 383; Ulmer 1985, Cf.; Stanley 2022a, p. 65). Derrida's definition of writing was also quite expansive and included the biological interest in "the most elementary processes of information within the living cell" (Derrida [1967] 1978, p. 9). For Derrida, writing "designates not only the physical gestures of literal pictographic or ideographic inscription, but also the totality of what makes it possible" (Derrida [1967] 1978, p. 9). For these reasons, Derrida suggested that the "entire field covered by the cybernetic *program* will be the field of writing" (Derrida [1967] 1978, p. 9). If grammatology was to be applied to biological

sciences, it would confront their own metaphysical assumptions. Derrida's argument is as follows:

> If the theory of cybernetics is by itself to oust all metaphysical concepts—including the concept of the soul, of life, of value, of choice, of memory—which until recently served to separate the machine from man, it must conserve the notion of writing, trace, grammè [written mark], or grapheme. (Derrida [1967] 1978, p. 9)

As commentators have noted, it can seem strange that the critique of metaphysics tends "to emerge when philosophers discuss the topic of writing" (Braver 2007, p. 436). However, recent work in biodeconstruction defends these interests against Malabou's unfounded concern that Derrida's comments were inconsistent with recent scientific work (Vitale 2018, pp. 73 and 213n32, Cf.; Malabou 2010, pp. 57–59). As Goldgaber also notes, even if Derrida did not develop grammatology as a science, he recommended its application to cybernetics as noted above (Goldgaber 2020, p. x).

This connection between Derrida's grammatology and cybernetic research into "man-machine hybrids" (Clark 2003, pp. 13–14) can similarly be seen in his interest in André Leroi-Gourhan's 1964 *Gesture and Speech* (Leroi-Gourhan [1964] 1993). As Malafouris and Renfrew note in *The Cognitive Life of Things*, Leroi-Gourhan anticipated "many subsequent philosophical arguments on the extended and distributed character of human cognition" (Malafouris and Renfrew 2010, p. 2). For Derrida, Leroi-Gourhan's contribution could be seen in his identification of a program in the evolution of life. As already noted, this program opened it up to analysis in the cybernetic sense. However, "cybernetics is itself intelligible only in terms of a history of the possibilities of the trace as the unity of a double movement of protention and retention" (Derrida [1967] 1998, p. 84). Leroi-Gourhan's challenge was to study how the "operational synergy of tool and gesture presupposes the existence of a memory in which the behavior program is stored" (Leroi-Gourhan [1964] 1993, p. 237). This was not limited to human beings and applied to all evolving living things, from a crab's claw to the present-day use of machines. As Derrida summarized, this program applied to the "'genetic inscription' and the 'short programmatic chains' regulating the behavior of the amoeba or the annelid up to the passage beyond alphabetic writing to the orders of the logos and of certain *homo sapiens*" (Derrida [1967] 1998, p. 84). In each case, a "general concept of the *grammé*" was implied. In this way, Derrida arrives at the logic of a general trace with biological implications (Derrida [1967] 1998, p. 84).

Like Ricoeur, Derrida also focused on Plato's denigration of the "evil of writing" (Derrida [1967] 1998, p. 34; Ricoeur 2004, p. 141; Plato 1995, p. 565). However, in "Plato's Pharmacy", Derrida was much more focused on how the generality of the trace could be seen in Plato's opposition between writing and thought, his substitution of "the prosthesis for the organ" (Derrida [1969] 1983, p. 108). Plato's critique of writing's exteriority was not only betrayed by its preservation in a written text. As Derrida pointed out, his argument was that memory is already touched by written exteriority. "Plato maintains both the exteriority of writing and its power of maleficent penetration, its ability to affect or infect what lies deepest inside" (Derrida [1969] 1983, p. 110). Writing is both exterior as well as interior, and that is its danger as far as Plato is concerned. However, precisely here, a space "opened up in the violent movement of this surrogation, in the difference between *mneme* and *hypomnesis*" (Derrida [1969] 1983, p. 109). Between memory and writing then, there is already "the space of writing, space *as* writing" (Derrida [1969] 1983, p. 109). It is in this sense that Derrida understands the trace as absence because the space also operates as a trace that "increases itself in the act of disappearing" (Derrida [1969] 1983, p. 110). Derrida's general trace thus emphasized the connection between memory and writing. "Writing and speech have thus become two different species, or values, of the trace" (Derrida [1969] 1983, p. 152). Written and psychical linguistic traces are thus central in Derrida's account, and

this results in his account of its general applicability to biology. However, at this point, he is not considering the cortical trace. Rather, the trace concerns inner speech and writing, two other senses of the trace noted in Ricoeur's account above and in EM.

When Goldgaber defends the plastic trace, she promotes a new materialist legacy for Derrida's grammatology. For these reasons, she also recognizes the trace's applicability to deciphering information in tree rings as well as "mnemonic properties of everyday objects" (Goldgaber 2020, pp. 142–43). For instance, MIT researchers demonstrated that "when sound 'hits' an object, the object vibrates, and the motion of this vibration creates a visual signal usually invisible to the naked eye" (Goldgaber 2020, p. 143, Cf.; Davis 2014). The idea that potentially any form may hold information does not reduce the object to readable text. Rather, it returns us to Derrida's account of writing as just "one of the representatives of the trace in general" (Derrida [1967] 1998, p. 167). As Derrida put it in *Writing and Difference*, "the concepts of general writing can be *read* only on the condition that they be deported, shifted outside the symmetrical alternatives from which, however, they seem to be taken, and in which, after a fashion, they must also remain" (Derrida [1967] 1978, p. 272). However, this is not to say that the trace in general "has little or nothing to do with the (anthropological, subjective, and so on) act of writing", as some early commentators surmised (Gasché 1986, p. 274). Rather, it is to note the persistent contact between Derrida's trace and its various instantiations.

The trace may be visible in writing, but that is not to say it is limited to that instantiation (Goldgaber 2020, p. 146). "To describe texts and matter grammatologically is to make explicit the intrinsic modifiability and retentiveness of the trace structure" (Goldgaber 2020, p. 146). This structure apprehends the multiple potentialities for representation and interpretation in the scientific research of material artifacts. Derrida was very clear that his use of the concept of the trace aimed to communicate with other "vulgar concepts" of writing (Derrida [1967] 1998, p. 56). In so doing, he did not seek to refound the sciences with an originary moment of writing. Rather, he provided a logic through which to ensure that scientific research remains open to the analysis of the trace across all its potential differences.

## 6. An-Other Trace

My contention is that this more recent work on the biological and material implications of Derrida's legacy remains relevant to EM. There are points at which Derrida's account of the trace is difficult, abstract, and can seem irrelevant to scientific interest in cortical traces. His prose is sometimes depicted as a "Pickwickian", intent to confuse his readers by those sympathetic to EM (Tallis 1995, p. 220). Ricoeur was also critical at times of Derrida's neologism *différance*, even if Derrida persistently tried to demonstrate their common interests (Derrida 2010, p. 173). As he noted in his debate with Ricoeur, "I have also attempted a critique of semiology" (Derrida 2010, p. 172). However, he "tried to keep us from forgetting that there are still signs in discourse, that discourse exists with the sign, with the differential chain, with spacing, etc." (Derrida 2010, p. 172). In the end, Ricoeur commended grammatology's importance, even if he may not have appreciated its full relevance (Ricoeur 2004, p. 139). My contention is that Derrida did not aim to escape Ricoeur's paradox so much as delineate a fourth trace that must be at work throughout the other three as written, psychic, and cortical. In part, this is because Derrida's other trace arises through similar discourses and themes found in Plato and Heidegger. It is also due to the nature of Derrida's deconstruction, which did not aim to surpass past philosophical projects so much as to demonstrate their interstices (Derrida and Caputo 1997, p. 74, Cf.; Stanley 2017, p. 17).

Nonetheless, there is a clear contrast between Ricoeur and Derrida. As Ricoeur put it, "now, in the trace, there is no otherness, no absence. Everything is positivity and presence"

(Ricoeur 2004, p. 426, citing; Changeux and Ricoeur 2000, p. 150). Whereas for Derrida, "the trace *is nothing*, it is not an entity, it exceeds the question *What is?* and contingently makes it possible" (Derrida [1967] 1998, p. 75). This fourth general trace for Derrida is the play of spaces itself in their differences which cannot be "restricted to the semiological element" (Derrida 2010, p. 173). "*The pure trace is differance.* It does not depend on any sensible plenitude, audible or visible, phonic or graphic. It is, on the contrary, the condition for such a plenitude" (Derrida [1967] 1998, p. 62). Ricoeur's trace addresses its presence. Derrida's trace provides the condition for its other instantiations. Derrida is careful to avoid reconstituting a metaphysical foundation. Hence, the trace can only arise in the play of its differences.

This is another way to explain why Derrida relentlessly critiqued "logocentrism" (Derrida [1967] 1978, p. 3). As Derrida commented elsewhere, "realism, sensualism—'empiricism'—are modifications of logocentrism" (Derrida 2004, pp. 64–65). He cited numerous examples of this philosophical preference for inner mental experience and the immediacy of speech (Derrida [1967] 1978, p. 12). For instance, he rejected Aristotle's preference for "mental experiences" that are translated into public language in *On Interpretation* (Derrida [1967] 1978, p. 11, Cf.; Aristotle 1938, p. 114). He went on to connect Aristotle and Plato's opposition between inner speech and external writing to the structuralist linguistics of Ferdinand de Saussure's similar preference (Derrida [1967] 1998, p. 30). For Saussure, "the spoken forms alone constitutes the object" (de Saussure [1916] 1974, p. 24, cited in; Derrida [1967] 1998, p. 31). Derrida also critiqued psychoanalytic interpretations of Jean-Jacques Rousseau's mind behind his written texts (Derrida [1967] 1998, pp. 158–59). This concern with psychoanalytic interiority was evident in Derrida's essay on "Freud and the Scene of Writing", which complemented *Of Grammatology.* Derrida took issue with Freud's "The Mystic Writing Pad", where Freud analogized the psyche to a slab of wax that can be endlessly written upon. As Derrida highlighted, Freud's psyche included "the possibility of this machine, which, in the world, has at least begun to *resemble* memory, and increasingly resembles it more closely" (Derrida [1967] 1978, p. 228). For Derrida, what psychology often presents as a more real inner mind or experience turns out to be reliant on exterior techniques and forms of writing. This is not an early theory of EM, but it is very near to Clark's interest in overcoming a "western prejudice" for the inner over the outer (Clark 2003, p. 26). Grammatology similarly aims to reframe this prejudice, but it does so with more direct references to the trace.

When Clark cites Heidegger's example of "transparent" tool use as a case of EM in *Natural-Born Cyborgs*, he defends a model of cognition that extends beyond skin and skull (Clark 2003, p. 48). However, as Derrida pointed out, the concept of extension (*Erstreckung*) itself includes a difference between two different senses of the tool's use (Derrida 2016, p. 147). As Heidegger described it, the human being "*stretches along* [*Erstreckung*] between birth and death" (Heidegger 1962, p. 425/373). Heidegger's challenge, as Ricoeur similarly noted with reference to *Gewesenheit*, is that empirical investigations (*Vorhandhenheit*) do not provide access to this aspect of human experience. Heidegger recalled that Descartes saw "*extensio* as basically definitive ontologically" (Heidegger 1962, p. 122/89). For these reasons, Descartes's concept of extension was of no use in Heidegger's aim to analyze human tool use as "ready-to-hand (*Zuhandenheit*)" (Heidegger 1962, p. 133/99). Here again, Derrida argued that, for Heidegger, "this extension is nothing, then; it is merely the empirical and fallen and inessential multiplicity of a presence, of a persistent *Vorhandenheit*" (Derrida 2016, p. 147). Whereas Ricoeur embraced the trace's presence and absence, Derrida outlined how a fourth general trace was also at work (Derrida 2016, p. 151). It does not matter whether the written, linguistic, or cortical trace is in view. Heidegger's contribution is that our experience is characterized by something other than the empirical physical nature of each case. After Derrida, the extended, traced nature of Heidegger's account of the tool operates as a

condition. The trace maintains contact with the various instantiations where that term applies. When the later Heidegger crossed being [*Sein*] out, he only reiterated the trace in Derrida's view (Derrida 2016, p. 224, Cf.; Heidegger 1958, p. 83).

Habermas's communicative action also orients Clark's interests with Derrida's similar irenic pursuits in democratic societies (Clark 1996, pp. 122–23, citing; Habermas 1990, pp. 58, 59, 134, 145). Habermas was critical of what he perceived to be Derrida's Heideggerian legacy and his persistent attention to religious terminology (Gordon 2015, p. 127, Cf.; Habermas 1987). Early in his career, Habermas did not recognize Derrida's profound critique of metaphysical and religious concepts in Heidegger's legacy. Nonetheless, the connection between Derrida's grammatology and the critique of the "metaphysics of presence" is now more regularly cited (Braver 2007, p. 436, Cf.; Stanley 2010, 2017, 2022a). Despite their differences, Derrida co-signed a newspaper article with Habermas explaining that their "aspirations converge regarding the future of the institutions of international law and the new challenges for Europe" (Habermas and Derrida [2003] 2006, p. 270). In the end, their agreement was more important in the face of rising violence, which they both recognized held religious aspects that called for renewed attention and institutional responses. What has since become clearer are the particularities and difficulties that arise in the challenge of religious strife (Stanley 2022b).

In sum, Clark provides a cognitive science complement to the deliberative communication between people of differing religious and moral viewpoints. This, in the end, may be his most salient contribution to the study of religion. However, here again, clarity about Clark's complementarity remains crucial. Language directly enhances our capacity to adjudicate moral and religious disagreements. However, in so doing, Clark presumes inherent openness to deliberation and willingness to exchange viewpoints. Moreover, in defending EM on this point, he suggests that the reasons for supporting this view could not be derived from cognitive science. My contention is that the trace opens EM to wider hermeneutic, political, and ethical justifications, some of which are already evident in Derrida's various evaluations of religion's multiple possibilities. Following Immanuel Kant, he understood "two strata" or layers of religion, one cultic and the other moral (Derrida [1998] 2002, p. 49, Cf.; Kant [1792] 1934, Book I, scts 3–4). Derrida recognized that a notion of interiority was at work in both strata and was not interested in a naive return to the Enlightenment era after the horrors of the twentieth century (Derrida [1998] 2002, p. 89). Nonetheless, in their difference, Derrida again noted a potential "trace" of religion that can only "begin and begin again: quasi-automatically, mechanically, machine-like, spontaneously" (Derrida [1998] 2002, p. 57). As I have written elsewhere, this trace encompassed religion's written materiality (Stanley 2022a, pp. 135–38). EM now presents another model to apply Derrida's trace to the future of situated religious cognition.

**Funding:** This research did not receive external funding.

**Institutional Review Board Statement:** Not applicable.

**Informed Consent Statement:** Not applicable.

**Data Availability Statement:** Not applicable.

**Conflicts of Interest:** The author declares no conflict of interest.

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
