# Peer review of "Tracing the Extended Mind"

_religions, doi:10.3390/rel16020189_

Round 1

Reviewer 1 Report

Comments and Suggestions for Authors

This article provides an important contribution to the discussion of theory of mind. It uses Philosophical, Religious and Cognitive psychological frameworks to engage the theory of the extended mind. The only question I have with this article is why is Jung not engaged at all in this discussion? Does the author see his contribution as irrelevant to the study? If so, perhaps it should be mentioned and give a short explanation. This is the only gap that I find in the argument. Otherwise, it is well written and the author engaged widely, specifically in the engaging with Derrida and Ricoeur.

Author Response

Comment 1: "The only question I have with this article is why is Jung not engaged at all in this discussion? Does the author see his contribution as irrelevant to the study? If so, perhaps it should be mentioned and give a short explanation."

Reply: I have endeavored to focus the essay on the range of figures covered and the literature that arose directly from each. So, insofar as Jung didn't arise for Clark, nor in Ricoeur or Derrida, I chose not to broaden my argument to cover his potential contribution accordingly. However, I plan to write further on this topic and will research the potential relevance of Jung’s thought for situated cognition, given it seems to be a potential gap for consideration.

Reviewer 2 Report

Comments and Suggestions for Authors

See my comments to the author attached here.

Author Response

Comment 1: "84: a critical point: Clarke’s position against the traditional philosophical dualism of ‘mind’ as distinct from the material ‘brain’: arguing for fusion. Here a note could be added to this traditional view."

Response 1: I don't think fusion is Clark's point nor my interest and I clearly stated that “establishing an identity between brain and inner speech is a moot point” (line 75). So I've tried to clarify throughout that my argument is focused on how best to articulate the correspondence and complementarity. I think Clark's quote here about the "western prejudice" adequately apprehends dualism and is also already strong enough (line 85).

Comment 2: 100ff "three aspects should be differentiated: (a) the mind or inner consciousness, (b) language, and (c) the biological brain."

Response 2: I've clarified that consciousness and mental representations are different, but that Clark’s focus is on the internalism that persists in Fodor and Churchland’s account of mentalese or neuralese. I added a few lines to this effect at 106ff and 119-120, where I cite Clark’s own intentional ambiguity on this point. I’ve also clarified at 128ff that Clark is considering the biological brain in his account of the mangrove effect. So, I think the consciousness, mind, brain distinction is more clear at this point in the paper. I would also add that I think this tripartite distinction is not always clear in Clark and the rest of the literature and this topic might deserve a separate treatment beyond the scope of this essay.

Comment 3: "128ff.: The transition of issues of morals and ethics should be explained. How does this passage relate to the previous? The implied equation of ‘morals’ with ‘problem solving’ appears as questionable. It may need to be explained."

Response 3: I've explained the transition a bit more and how problem-solving matters in this context of moral diversity at the transition to that paragraph now at line 135, and at the end of that paragraph in lines 160ff.

Comment 4: "155ff.: How then to ´come from ‘morals’ to the ‘religious’? The case presented here depicts specific Islamic codes of conduct or mores, but no the ‘religious’ as such, as relating to the divine(transcendent). It is a about a specific religious system of mores, and might be left under the headingor ‘morals’ instead of ‘religion’."

Response 4: I've explained a bit more at this point on the distinctiveness of religious language and texts in the literature I've written about elsewhere. I’ve also expanded more on translation with context at lines 180-197 in the revised draft.

Comment 5: "163ff.: Comment: Here two new themes are introduced. (a) that language is historical, and stores cultural memory, inevitably and essentially, and (b) the issue of intercultural communication (as between different languages, with issues of translation. At this point it might be advised to differentiate between ‘language’ as such, ideal-typical, and concrete specific languages. In which form ‘language’ exists concretely. This should be made explicit. Comments 166ff, 188ff, 199, 204ff, 227ff and 271ff all have to do with this same issue of linguistic diversity. 

Response 5: I have expanded a bit on different languages, translation matters and linguistic diversity in particular in lines 180-197. I am also noting here that I noted Carruthers' response to Whorfian relativismm and have expanded that a bit on line 68. Essentially, I think I may just have to note that I disagree with the reviewers assumptions here but think that the literature I've cited justifies my approach in the paper and for further debate in the field.

Comment 6: 286ff... "The reference to the ‘trace’ is fine. But the step from ‘language’ to ‘biology’ is not shown convincingly... The interconnection between ‘language’ (in its specific instance) and the ‘brain’ does not support a claim to their assumed identity."

Response 6: I have clarified a bit more on the relation between language and biology in Clark’s discussion of mangroves above. That essay specifically includes the brain in a few places and I’ve added a lines 125-127 in the revised draft. I also would like to reiterate, that in the final two paragraphs of this section and lines the reviewer is referring to, I have already stated that an identity is not assumed by Clark nor myself. On line 325-26 in the revised draft, I state that Clark does not think there is a direct identity between brain and language. When he develops concepts like hybridity concepts like the trace also emerge. The vehicle/content distinction is also crucial to this point, so I've focused on that in conclusion and forecasting the next two sections on lines 350-52 in the revised draft. This is also more clear in subsequent sections, where the reviewer seems to have understood better in their and provided more positive comments.

Comment 7: "383...The term ‘correspondence’ should be noted, which implies relationship, also similarity, but not identity. The ‘phenomena’ may be external and internal as to how the phenomena present themselves to minds and perceptual faculties shaped by specific languages."

Response 7: I've tried to make this a bit more clear here at lines 361-63 in the revised draft.

Comment 8: "Interesting to identify the ‘trace’ here, as metaphor, in the relation of mind and brain. Which connects to the above made points of the specific nature of this relation according to specific languages ‘in concreto’.

Response 8: My sense is that the reviewer began to see this here, and that as I've clarified a few things above, the argument started to hold for them better by this point.

Comment 9: 411... Here the key word of ‘semiotics’ emerges. It could be emphasised. What follows is a structuralist critique of ideas of immediate representation, as unconditioned by ‘language’."

Response 9: I do not think semiotics necessarily results in structuralist critique. I've added a few notes in subsequent sections on Ricoeur and Derrida's debate in the conclusion to help clarify this point at lines 684 and response 12 below.

Comment 10: 451ff "The brief excursus on Heidegger introduces the ideas of practical memory and of the life- world as contexts of ‘memorialisation’ ss further horizons that could be explored, beyond other brief mention here."

Response 10: The essay is already quite long and I think this is beyond the scope as I'm citing Heidegger to provide context and because it connects to Clark's similar interest to Ricoeur.

Comment 11: 599 "This contradicts the above to some degree, but unfolds a side aspect, by exploring the imprint of language on the brain"

Response 11: I think this is a complex point, but I don't think that this conclusion contradicts the above. The issue is further scientific research might find that imprints on the brain occur. But that remains an open matter. The plastic trace here aims to keep the science open to further inquiry. That is made clear in this paragraph, which I’ve broken up a bit to help the reader may follow the point more clearly. I also make clear on line 433 in the draft that the brain scan doesn’t change the problem of representational interpretation. This is the context of Ricoeur’s debate with Changeux but I’ll trust the reader to make the connection. In any case, I think this section is clear that the trace maintains the need to keep pursuing new neuroscientific research on this matter.

Comment 12: "625... This ignores the essentially structuralistic conception of Derrida. The introduction of his concept of ‘trace’ is situated herein. Thus. it may rather be the character of the specific language, and the trances of the forces which shape it, that are an intermediate step. From here, the formative influence of ‘language’ as such, and concretely, of specific languages the brain, is and fruitful. To claim it to be a ‘Materialistic reading’ of Derrida is, in my view not fully sustainable."

Response 12: Similar to response 9 above, I do not think that a structuralist reading is implied by semiotics, necessarily. I have expanded a bit on Ricoeur's debate with Derrida to explain this a bit more (lines 656-59). I've also added a line on Derrida’s critique of Saussure’s preference for speech in lines 684-88 in the paragraph on logocentrism. The new materialist interpretations of Derrida aim to approach prior structuralist/post-structuralist debates in new ways. I don’t think I can do more in this context to sustain those materialist readings. My contribution here is to connect them to EM literature. I’ll keep this in mind though for future research or follow up projects on this matter.

Comment 13: 632ff "Is reference to the ‘psychic’ is made, the work of Jacques Lacan could be included here. He reflects on the relation between ‘language’ and the ‘psychic’ of the desires and the realm of bod and soul, Here, ‘difference’ exists in relation to what remains beyond languages, but motivates its meanings.

Response: Lacan is beyond the scope of this paper. Potentially, future work might be interesting on this aspect, though.